# Multiple Functions of Malpighian Tubules in Insects: A Review

**DOI:** 10.3390/insects13111001

**Published:** 2022-10-31

**Authors:** Priscilla Farina, Stefano Bedini, Barbara Conti

**Affiliations:** Department of Agriculture, Food and Environment, University of Pisa, Via del Borghetto 80, 56126 Pisa, PI, Italy

**Keywords:** bioluminescence, brochosome, cocoon, excretion, foam, inorganic salt, mucofibrils, mucopolysaccharide, protein, silk-like fiber

## Abstract

**Simple Summary:**

The Malpighian Tubules (MTs) are well known and studied as the main excretory organs in most insects. However, MTs, despite their quite simple morphology, can also serve numerous specialized functions in some species such as the production, processing, and storage of mucopolysaccharides, proteins, mucofibrils, adhesive secretions, brochosomes, silk-like fibers, and inorganic salts as well as the remarkable phenomenon of bioluminescence. In this review, we attempted to summarize the observations and experiments made from the 1850s to the present day concerning the non-excretive functions of insects’ MTs, underlying, when necessary, the need for new investigations supported by the numerous technologies currently available (e.g., chromatography, spectroscopy, microscopy, proteomic, genomic) to validate outdated theories and clarify some dubious aspects.

**Abstract:**

The Malpighian Tubules (MTs) are the main excretory organs in most insects. They play a key role in the production of primary urine and osmoregulation, selectively reabsorbing water, ions, and solutes. Besides these functions conserved in most insects, MTs can serve some specialized tasks at different stages of some species’ development. The specialized functions include the synthesis of mucopolysaccharides and proteins for the building of foam nests, mucofibrils for the construction of dwelling tubes, adhesive secretions to help the locomotion, and brochosomes for protection as well as the usage of inorganic salts to harden the puparia, eggs chorion, and pupal cells’ closing lids. MTs are also the organs responsible for the astonishing bioluminescence of some Diptera glowworms and can go through some drastic histological changes to produce a silk-like fiber utilized to spin cocoons. The specialized functions are associated with modifications of cells within the entire tubules, in specific segments, or, more rarely, modified secretory cells scattered along the MTs. In this review, we attempted to summarize the observations and experiments made over more than a century concerning the non-excretive functions of insects’ MTs, underlying the need for new investigations supported by the current, advanced technologies available to validate outdated theories and clarify some dubious aspects.

## 1. Introduction

The Malpighian Tubules (MTs) are named after the Italian biologist and physician Marcello Malpighi, their first discoverer in 1669. In most insects, MTs are blind-ended tubules that freely lie in the hemocoel, emptying their secretions into the gut at the midgut-hindgut junction. They show a great diversity in number (from two to more than 250), shape, length, diameter, morphology, functional segmentation, arrangement, and histology among species [1,2,3,4]. Generally, there is an inverse relationship between the number and length of MTs. Indeed, longer MTs are less numerous and vice versa, given that the filtering surface per mg of insect remains roughly the same (e.g., 412 mm^2^/mg in *Periplaneta* sp. which has 60 MTs and 500 mm^2^/mg in *Gastropacha* sp. which has only six MTs) [1].

From a cytological point of view, MTs typically consist of epithelial tissue made of two cell types: the principal cells deriving from the ectoderm and the secondary cells from the mesoderm [2,4]. The principal cells (Figure 1), five times more numerous than the others, have a brush inner border rich in slender microvilli containing a core of microfilaments and digitate extensions of the numerous mitochondria and an extensive endoplasmic reticulum on the lumen side. The function of the microvilli is to increase the surface area between the cell and the lumen of the tubule, enhancing the capacity for ion and fluid transport. The principal cells are, indeed, active in the transport of catabolites as well as hydrogen, potassium, and sodium ions and are involved in the primary urine formation [3,4]. The secondary cells, implicated in the chloride flow and primary urine modification, have aquaporins and small microvilli without mitochondria [2,4].

MTs are the main excretory organs in the classes of Insecta, Arachnida, and Myriapoda (Arthropoda) and in Tardigrada. Among insects, the MTs system is rudimental in Strepsiptera and absent in Collembola and Aphidoidea. In springtails, the water uptake and ion transport are carried out, in exchange, by the collophore [6], while in aphids the key osmoregulatory role relies on the gut [7].

The main task of MTs is to purify the hemolymph by removing waste metabolites and toxins and maintaining a proper balance among organic solutes and water [8]. MTs are highly permeable to low-molecular-weight molecules, so they have a pivotal role in actively uptaking and filtering catabolites, solutes (e.g., amino acids, sugars, ions), and water from the surrounding hemolymph, generating the so-called primary urine [2,3]. The soluble compounds and water necessary for the insect are, in the end, reabsorbed in the proximal segment of the MTs, in the rectum, or ileum, maintaining constant the internal osmotic pressure [2,9]. As excretory organs, MTs are considered analogous to the vertebrate nephrons and express homologous human genes. Indeed, several human kidney disorders are currently studied on the MTs of the genetic model insect *Drosophila melanogaster* Meigen, 1830 (Diptera: Drosophilidae) [10]. 

The toxic nitrogenous waste solutes, derived from proteins and purines metabolism, can be converted and eliminated by the MTs as uric acid, allantoin, allantoic acid, urea, or ammonia, depending on the enzymes and biochemical pathway involved, which, in turn, are related to the insect diet and ecological niche [2,11]. Besides nitrogenous waste, MTs can also actively excrete toxic elements of natural and chemical origin (xenobiotics) to let the insects better adapt to a wide range of nourishment. Furthermore, they express high levels of enzymes such as alcohol dehydrogenases, several cytochrome P450s monooxygenases (mainly belonging to the insect-specific class Cyp6), and glutathione-S-transferases which play a key role in the metabolism and detoxification of such substances [9,12,13,14,15,16,17]. Among the natural toxic compounds metabolized, we can list the alkaloids atropine, morphine [18,19], capsaicin, dihydrocapsaicin [20], and nicotine [18,21,22] and the secondary metabolite salicylate [23]. Furthermore, MTs can also mitigate the toxic activity of insecticides such as dichloro-diphenyl-trichloroethane (DDT) [24], lufenuron [25], and permethrin [26], leading to the serious issue of resistance development.

In *D. melanogaster*, and likely in other Diptera, MTs also have a protective role in the “epithelial immunity”, as they sense the presence of bacterial infections and secrete several anti-microbial peptides (typically deriving from the fat body and hemocytes) constituting a cell-autonomous immune system [27,28]. Lysozyme and lectin, two antimicrobial peptides, were also identified through shotgun proteomics in the MTs of fifth instar larvae of *Bombyx mori* Linnaeus, 1758 (Lepidoptera: Bombycidae) [29].

However, MTs, despite their quite simple morphology, can also serve numerous specialized functions related to modifications of entire tubules, specific segments, or, more rarely, of modified secretory cells scattered along the MTs. Some examples are the production, processing, and storage of mucopolysaccharides, proteins, mucofibrils, adhesive secretions, brochosomes, and inorganic salts and the remarkable phenomenon of bioluminescence. Another specialized function of the MTs, carried out after going through drastic histological changes, is the secretion of a silk-like fiber for the construction of cocoons. In this review, we attempted to summarize the observations and experiments made over more than a century concerning the non-excretory functions of insects’ MTs (Table 1), to provide, as much as possible, a detailed and rather complex picture of these multifunctional organs. Unfortunately, some of the specialized functions here reported were described only in outdated studies that should be supported by new investigations carried out exploiting the current, advanced technologies available.

**Table 1 insects-13-01001-t001:** Specialized functions related to the Malpighian Tubules.

Product	Function	Taxonomy	References
**Mucopolysaccharides and proteins**	Construction of foam nests	Rhynchota: Homoptera: Aphrophoridae, Cercopidae	[30,31,32,33,34,35,36,37,38,39,40,41,42,43,44,45,46]
**Mucofibrils**	Construction of dwelling tubes	Rhynchota: Homoptera: Clastopteridae, Machaerotidae	[35,47,48,49,50,51,52,53]
**Adhesive secretions**	Construction of oothecae	Coleoptera: Chrysomelidae	[54,55]
	Facilitation of locomotion	Coleoptera: Chrysomelidae Neuroptera: Chrysopidae	[56,57,58,59]
**Brochosomes**	Protection of eggs, juveniles, and adults	Rhynchota: Homoptera: Cicadellidae (ascertained), Psylloidea Heteroptera: Berytidae, Lygaeidae, Miridae, Plataspidae, Rhopalidae, Saldidae, Tingidae Diptera: Culicidae	[8,15,60,61,62,63,64,65,66,67,68,69,70,71,72,73,74,75,76,77,78,79,80,81,82,83,84,85]
**Other organic secretions**			
Carbohydrates and proteins	Building of hallways and chambers	Rhynchota: Homoptera: Cicadidae	[43,86,87,88,89]
Glycoproteins and lipoproteins	Reinforcing of shields	Rhynchota: Homoptera: Coccoidea	[90,91,92]
Glycosaminoglycans, polysaccharides, and proteins	Unclear (related to cocoons covering)	Hymenoptera: Apidae, Megachilidae	[93,94,95,96,97]
**Inorganic salts**			
Calcium (calcospherites and granules)	Hardening of puparium	Diptera: Muscidae, Tephritidae	[98,99,100,101,102,103,104,105,106]
Calcium (lime glands)	Precipitation of excessive salts	Diptera: Ephydridae	[107]
Calcium and phosphorus	Acquisition of freeze tolerance	Diptera: Tephritidae	[108]
Calcium	Hardening of eggs chorion	Phasmatodea: Heteronemiidae	[109,110,111,112,113,114]
Calcium	Hardening of cocoons	Lepidoptera: Erebidae, Lasiocampidae	[115,116,117,118,119]
Calcium	Construction and closure of pupal cells	Coleoptera: Cerambycidae	[88,120,121,122]
**Bioluminescence**	Attraction of prey and partners	Diptera: Keroplatidae	[16,123,124,125,126,127,128,129,130,131,132,133,134]
**Silk-like fibers**	Construction of cocoons	Coleoptera: Carabidae, Curculionidae (: Hyperinae)	[135,136,137,138,139]
	Construction of cocoons	Hymenoptera: Eulophidae	[140,141,142,143,144]
	Adhesive for debris	Neuroptera: Chrysopidae	[145,146]
	Construction of cocoons	Neuroptera: Ascalaphidae, Chrysopidae, Coniopterygidae, Myrmeleontidae, Sisyridae	[58,147,148,149,150,151,152,153,154,155,156,157,158]
	Construction of cocoons	Thysanoptera: Aeolothripidae, Melanthripidae, Thripidae	[159,160,161,162,163,164,165,166]

## 2. Specialized Functions of Malpighian Tubules

### 2.1. Mucopolysaccharides and Proteins 

A specialized function performed by the MTs is the synthesis of compounds involved in the creation of the “foam nest” where juveniles belonging to the Homoptera Aphrophoridae and Cercopidae families spend their whole preimaginal lives. Due to the production of this foam that looks like a spittle mass (Figure 2a), these species are commonly known as “true spittlebugs”.

In 1912, Licent [30] assessed, for the first time, that the foam is composed of the honeydew produced by the filter chamber and a large amount of mucopolysaccharide and proteinaceous substances acting as a surfactant [30]. About 50 years later, Marshall [34] performed a specific histological study on several true spittlebugs’ MTs showing that, in the preimaginal instars of the subfamilies Aphrophorinae and Cercopinae, the four MTs are divided into three parts: a smooth proximal segment, a lobulated distal segment, and a slim duct. The proximal segment carries out the synthesis and excretion of the mucopolysaccharides to form the foam, while the distal segment secretes the proteinaceous substances. 

In the proximal segment (Figure 2f), the cells are large, with large, oval nuclei (sometimes stellate) and numerous granules oriented into stacks, and they contain the vacuolated mucocomplex. The disappearance of the granules coincides with the vacuolated mucocomplex appearance, which suggests a secretory cycle [34]. The mucopolysaccharide, blended with the abundant honeydew, decreases the surface tension to stabilize the froth. To “whip” the foam, the nymphs blow in air from the thoracic and abdominal stigmas and repeat movements of closure and separation of the paratergal lobes placed on their last abdominal sclerites (Figure 2b). 

The protein synthesis takes place in the distal segment (Figure 2f) that contains several membrane-bounded granules, numerous Golgi bodies, vesicles, canaliculi lined with microvilli, and a broadly developed rough endoplasmic reticulum. Such structure is characteristic of protein-secreting and -exporting cells [35,43]. The proteins here produced are then secreted into the MTs lumen, where they are used as surfactants to stabilize the foam [30]. The presence of peptides, polypeptides, and proteins in the spittlebugs’ froth has been confirmed by various authors over the years through several chemical methods, light and electron microscopy, and gel and automated electrophoresis [32,33,36,40,42]. Protein synthesis is considered a form of excretion in true spittlebugs’ juveniles and ceases at the adult stage [35,43]. The secretion of products with a similar chemical nature from the MTs is also reported in cicada juveniles [43,88], as discussed in Section 2.5.

According to Evans [31], true spittlebugs juveniles molt inside the froth to protect themselves from excessive evaporation. This theory, still widely accepted, was questioned by Turner [38], who evaluated the water evaporation from the foam. From its experiment on the aphrophorid *Aphrophora saratogensis* (Fitch, 1851), Turner [38] deduced that the protection against desiccation provided by the foam is just moderate. Moreover, most of the spittlebugs live in mesophilic habitats, and their diet on xylem sap ensures them unlimited access to water. Besides, Evans’ theory [31] does not explain why many cercopid hypogeal nymphs such as *Haematoloma dorsata* (Ahrens, 1812) (Figure 2c,d) produce the foam as well [37]. 

More recent studies attribute other functions to the foam. For example, in the case of the hypogeal cercopid *Mahanarva* (*Mahanarva*) *fimbriolata* (Stål, 1854) nymphs, the foam can maintain the temperature constant during the daily fluctuations and with a value similar (difference of ≤ 0.2 °C) to that of the soil. The stabilizing and modulating effect on the bubbles is due to the content of palmitic and stearic acids, carbohydrates, and proteins [45]. According to Chen et al. [44], in mesophilic epigean habitats, the foam protects the spittlebug nymphs from excessive UV radiation. Indeed, juveniles of the Cercopidae *Callitettix versicolor* (Fabricius, 1794) (Figure 2e) are lighter in color and thinner than adults, so they are more susceptible to radiation damages. The protection from UV is due to the gas-liquid interfaces among bubbles that reflect the incident light. The greater the concentration of the organic components in the foam, the smaller the bubbles, and the higher their optical effect [44]. The foam produced by the nymphal instars of the aphrophorid *Poophilus costalis* (Walker, 1851) is composed mainly of chitinases and proteases enzymes, with an antifungal effect against the soil-borne fungus *Fusarium oxysporum* f. *pisi* [46]. Similarly, the foam of *Aphrophora costalis* Matsumura, 1903 (now *Omalophora costalis* (Matsumura, 1903)) shows fungicidal activity and is, therefore, considered a potential weapon against entomopathogenic and phytopathogenic fungi [39]. A defensive role was also reported for the foam produced by the pine spittlebug *Aphrophora cribrata* Lethierry, 1890 (now *Epipyga cribrata* (Lethierry, 1890)), as it is repellent against the predatory ant *Formica exsectoides* Forel, 1886 (Hymenoptera: Formicidae), thanks to the presence of a mixture of metabolites probably deriving from the host plant *Pinus strobus* Linnaeus (Pinaceae) [41]. 

### 2.2. Mucofibrils 

The MTs of the Clastopteridae and Machaerotidae homopteran families belonging to the super-family Cercopoidea produce mucofibrils during their preimaginal life. These insects, living in tropical and subtropical regions of Europe, Africa, and Asia, are commonly known as “tube spittlebugs” because their nymphs build mineralized, rigid, roughly conical tubes in which they live immersed in a liquid excreted from the anus. Their dwelling tubes have species-specific shapes and are sealed to the stems of host plants, generally with the opening facing upward. Inside the tube, the cercopoid juveniles position themselves with the abdominal tip directed toward the opening, and, with the head downward, they can insert the stylets into the plant tissue and suck the sap [50].

Marshall [47,48] documented the histochemical changes occurring in sections of the MTs epithelium of several machaerotid nymphs during the production of both the froth, like that of the true spittlebugs, and the mucofibrils used for the construction of the dwelling tubes. Their four MTs consist of three distinct regions: a lobulated distal segment, five times longer than the adjacent smooth proximal segment, and a slim outlet duct. A similar regional differentiation in the MTs is also reported in true spittlebugs ([34]—Section 2.1) and cicadas ([88]—Section 2.5). The proximal segment can be divided into two further zones: an anterior one with fibril-rich cells (fibril zone) that occupies three-quarters of the total length of the proximal segment, and a posterior one with granules-rich cells (granule zone) (Figure 3).

The granule zone is responsible for foam production, and its cells have an extended rough endoplasmic reticulum and several Golgi bodies, typical of cells that synthesize and export proteins. The froth appearance coincides with the gradual vacuolation of this zone and happens only before each ecdysis [47,50]. The granule zone is cytologically comparable to the entire proximal segment of true spittlebugs’ MTs, but it is shorter [47]. In the anterior zone of the proximal segment are present protrusions of cytoplasmic material into the MT’s lumen and a coalesced matter made of mucofibrils. Mucofibrils contain glucosamine, glucose, glucuronic acid, and proteins identified through paper chromatography [49]; they are arranged in stacks, oriented radially to the tubule, and their size and shape differ according to the tube spittlebug species. The production of such geometrical bodies within MTs was compared to the brochosomes’ case [48], here discussed in Section 2.4. Mucofibrils are extruded at intervals into the lumen of the fibril zone, pass down the hindgut, and are poured out through the anus. Machaerotids build the meshwork of the tube executing circular or semi-circular abdominal movements around its lip [48]. The framework is made of mucofibrils on which are laid layers of organic materials impregnated with inorganic minerals. Calcium, manganese, magnesium, and phosphate derive from spherites secreted by a specialized region of the midgut (called region D), while ferrous iron and potassium come from a region named C. The tubes were examined through electron microscopy, micro and semi-micro chemical analysis, chromatography, and spectrophotometry [51]. Calcium protects and supports the structure of the dwelling tube; these are functions similar to the numerous examples of calcification reported in Section 2.6.

Unfortunately, despite the interest in the Clastopteridae and Machaerotidae families’ taxonomy and phylogeny [52,53], there have not been new works about their peculiar production of mucofibrils in the past 50 years. The quantification and qualification of the proteins produced by their MTs, perhaps through colorimetric assays, chromatography (e.g., HPLC, LC-MS/MS), spectrophotometry, fluorescently or radio-chemically labelled proteins, and electrophoresis, could help to reconstruct the molecular phylogenetic placement of the species on such a basis.

### 2.3. Adhesive Secretions 

In some Coleoptera Chrysomelidae leaf beetles, the MTs produce a sticky material then applied onto the eggs to create the oothecae. This specialized function is widely spread in the tribe Cassidini, such as in *Cassida* (*Cassida) nebulosa* Linnaeus, 1758 and *Cassida* (*Cassida*) *rubiginosa* Müller, 1776. Their eggs are not just bound to a vegetal surface only with the secretion of the female colleterial glands as typical in other chrysomelids, but they are also dipped in several layers of a brown-reddish MTs proctodeal secretion. Finally, the egg masses are further covered with a liquid fecal coating [54,55]. 

The larva of the chrysomelid beetle *Agelastica alni* (Linnaeus, 1758) (Figure 4) uses the last abdominal segment as a pseudopod and extrudes a sticky secretion from its anus to have more adhesion to the vegetation and ease its locomotion. The adhesive secretion is produced from the distal ends of the six MTs, which appear swollen, with large cells and nuclei, and with cytoplasm rich in granules and vacuoles [56]. 

Both these specialized functions in chrysomelids have never been further investigated in recent years; a chemical, genomic, and proteomic approach to characterize the secretions and some biological observations could be valid starting points.

Similarly, the first, second, and third larval instars of some Neuroptera Chrysopidae species use a sticky exudate made of proteins, produced by their eight MTs, to adhere to the substrate and help their progression. The color and viscosity of the adhesive exudate vary among the species: it is dark brown and thick in *Chrysopa (Chrysopa) carnea* Stephens, 1836 (now *Chrysoperla carnea* (Stephens, 1836)), pale yellow and fluid in *Chrysopa (Chrysopa) cubana* Hagen, 1861 (now *Ceraeochrysa cubana* (Hagen, 1861)), and copper red and thick in *Chrysopa (Chrysopa) rufilabris* Burmeister, 1839 (now *Chrysoperla rufilabris* (Burmeister, 1839)) [57]. In these species, the proximal ends of the MTs and the first third of their distal ends are swollen, and the cells display small, crystalline globules, small vacuoles, and large nuclei. The adhesive secretion can also be a strong defensive agent against ants [59]. Considering the proteinaceous nature of the adhesive substance, it might be the same silk-like material produced by the MTs when chrysopids pupate (see Section 2.8) or a precursor [57]. Furthermore, mature third instar larvae approaching pupation show a lower defensive ability [58], probably due to the reduced production of the adhesive substance in favor of the silk-like compound. Such hypotheses could be validated through modern, proteomic technologies.

### 2.4. Brochosomes

Brochosomes (from the Greek βρόχος = “mesh of a net” and σῶμα = “body”) [62] are peculiar structures produced by the MTs. They are ultramicroscopic (0.2 to 0.7 μm in diameter), polyhedral, honeycombed spheres, internally hollow (Figure 5a), containing lipids and a family of proteins [8,63] recently named “brochosomins” [85]. Their size, shape, surface configuration, number, and distribution can vary interspecifically, among sexes, or developmental stages [67]. The mucofibrils produced by the tube spittlebugs juveniles (see Section 2.2), even if a lot less intricate, are the only comparable bodies found, so far, within the MTs [48].

Brochosomes were firstly recorded from Diptera Culicidae and Homoptera Auchenorrhyncha specimens [61] and considered, for a long time, a prerogative of the Cicadellidae major subfamilies. A more recent work documented their presence, even if in smaller amounts, also in a Homoptera Psylloidea species and some species of the Heteroptera families Berytidae, Lygaeidae, Miridae, Plataspidae, Rhopalidae, Saldidae, and Tingidae [79]. However, not all authors agree with this report. According to Rakitov [74], the occurrence of brochosomes on non-cicadellid specimens could be due to samples’ contamination. 

In Cicadellidae leafhoppers nymphs and adults, brochosomes develop through four stages in Golgi-derived vesicles within the specialized, glandular, middle segment of their four MTs. This segment is rod-shaped, inflated, and thick, and its lumen is surrounded by a single layer of secretory cells with large, spherical nuclei and extended rough endoplasmic reticulum. The cytoplasm of the glandular segment is occupied by several vacuoles containing developing brochosomes (Figure 5b). The complete gradual maturation of brochosomes takes place in secondary vacuoles outside the Golgi bodies, and then they are discharged through exocytosis into the tubule lumen and pumped through the hindgut [15,69].

After each molt, leafhoppers excrete the newly developed brochosomes from the anus in a colloidal fluid medium (Figure 5c). As a first step, the fluid is smeared as a coat on immatures and adults, using the hind tibiae, and deposited on a specific “wax field” near the costal margin of the tegmina. This act is called “anointing” [68]. When the brochosomes are dry, they are further distributed over the integuments through acts of “grooming”, thanks to the modified hind and fore legs that have rows of specialized macrosetae on the tibiae [73,75,81]. In the end, brochosomes are mainly found around the antennae, on the head (eyes excluded), pronotum, thorax, abdomen, coxae, the lower and upper surface of wings, and male genitalia, with a species-specific distribution [67]. 

The function of brochosomes is still uncertain. According to different authors, they can act as protective elements against desiccation and temperature fluctuations and keep away soil particles, microbes, fungal spores, predators, parasites, water, liquid exudates (their own or from other specimens), and sticky substances produced by plants from the bodies [65,69,71,75,82,83,84]. They are also supposed to act as carriers of aggregation and sex pheromones [64], reflect UV radiation in case of excessive solar brightness, and provide thermal insulation [66].

The camelthorn gall leafhopper *Scenergates viridis* (Vilbaste, 1961) forms true galls transforming the leaves of the Fabaceae shrub *Alhagi maurorum* Medikus. This species uses brochosomes mixed with honeydew and crushed exuviae to plug the galls and brochosomes with wax platelets (produced by the host plant) to make the inner surface of the chamber hydrophobic and non-stick. The plugs represent mechanical barriers and sticky traps against potential intruders, while the inner covering helps to maintain the chamber free of excrement [80].

In several species of the Cicadellidae tribe Proconiini, besides the integumental brochosomes, egg brochosomes have also been identified. These structures are elongated, rod-shaped, up to 20 µm long, with intricate surface microsculptures [72]. Before the oviposition, the Proconiine females collect the brochosomes from the anus using their hindlegs and store them on the forewings, where they dry in the form of convex pellets [60]. Then the females brush the pellets with strokes of their hind tibiae, coating their egg clusters and the oviposition site with a white brochosomes powder. The secretory region of the MTs switches from the production of integumental brochosomes to egg brochosomes when females are developing eggs [70].

The brochosomes covering is supposed to protect the egg masses against parasitoids, predators, and pathogens and create optimal conditions for their development [71,72,77]. Egg brochosomes, considering their unique species-specific features, are suggested as helpful identification keys for some Proconiine species in *Citrus sinensis* (Linnaeus) Osbeck, 1765 (Rutaceae) orchards [76] and in vineyards [78], where these pests can be vectors of the dangerous bacterium *Xylella fastidiosa* (Xanthomonadaceae).

The continual research conducted by Rakitov and his colleagues from the nineties up to recent years [67,68,69,70,71,72,73,74,80,82,83,85] has examined in detail the structure, composition, and spreading of brochosomes and the role of MTs in their development.

### 2.5. Other Organic Secretions

Cicada nymphal stages live in cryptobiotic micro-habitats in the soil, feeding on xylem fluids from roots [167]. The four MTs of cicadas’ pre-imaginal instars can be divided into five sections: a thin anterior-most segment underlying the filter chamber, a short intermediate duct, a short, swollen, smooth proximal tract, a long, nodulose distal segment, and a thin terminal tract that ends in the rectum [89]. The proximal tract of the MTs is composed of large secretory cells producing granules of an acid mucopolysaccharide, while the distal segment displays cells with extensively developed rough endoplasmic reticulum and secretory vacuoles, producing a substance rich in proteins and carbohydrates [89]. As first observed by Fabre [86] in the cicada *Tibicen plebeius* Scopoli, 1763 (now *Lyristes plebejus* (Scopoli, 1763)), both the secretory products of the MTs mix with the copious watery excretion coming from the anus to create a fluid with mechanical and physiological properties. The proteinaceous component of the fluid is useful to agglutinate the soil particles and build the inner walls of their underground hallways and chambers and the aboveground “turrets” of exit (Figure 6) [86,87]. The fluid is also used to remove mud from legs and integument [87], buffer soil pH [88], and it is supposed to be fungicidal and fungistatic [89]. 

Despite the differences in their life habits, there are structural and functional similarities between the previously mentioned Cercopoidea (Section 2.1) and the Cicadoidea juveniles [43,89]. Similarly to cicadas, true spittlebugs nymphs produce a mucocomplex from the proximal segment of the MTs [34] and a proteinaceous secretion from the distal segment [35], and, in both groups, the secretion of such substances ceases in adults [43,89]. The evidence suggests that the proximal and distal segments of the MTs are homologous and that the individual adaptations of Cercopoidea and Cicadoidea species (foam nests vs. underground burrows) might have a common origin [89]. 

In scale insects (Homoptera: Coccoidea: Diaspididae), the cuticular glands cells on the dorsum secrete an abundant waxy secretion in the shape of long, white, double-stranded filaments [90]. This secretion is blended with a cementing glycoproteinaceous and lipoproteinaceous matrix produced by the MTs and deposited by the pygidium. In this case, the secretory cells are unexpectedly scattered along the MTs, among the regular excretory cells, without forming specialized, separated segments like in true spittlebugs, tube spittlebugs, and cicadas (see Section 2.1, Section 2.2, and Section 2.5). Once poured out, the filaments are used to build the characteristic shield mesh on the back and secured with the organic MTs product [91,92].

The exact content in carbohydrates, lipids, and proteins of the organic secretions produced by cicadas and scale insects’ MTs is still unknown but could be investigated through chemical and proteomic techniques.

The mature larvae of the Neotropical bumblebee *Bombus atratus* Franklin, 1913 (now *Bombus pauloensis* Friese, 1912) (Hymenoptera: Apidae) produce a filamentous secretion made of carboxylated and sulfated acid glycosaminoglycans, neutral polysaccharides, and proteins. The function of this mucus that fills the lumen of the MTs is still unclear: the secretion becomes evident when the larva starts spinning its cocoon, but it is not known for certain if it plays a role in the cocoon’s elaboration [93,94]. Further topochemical and microscopic investigations would be welcomed to clarify this doubt. A mucus with the same chemical composition is also produced by the MTs of the Apidae species *Plebeia droryana* (Friese, 1900) and *Scaptotrigona postica* (Latreille, 1807) [95]. More recent topochemical, polarized light microscopy, and microspectroscopy investigations reported that the cocoon’s wall built by the solitary Megachilidae bee *Lithurgus chrysurus* Fonscolombe, 1834 is made of layers of thin, twisted silk fibroin filaments distributed parallel to the surface [96]. In the front side, the wall composition is integrated with a layer of a similar white, opaque, fine-grained mucus produced by the MTs [97]. 

### 2.6. Inorganic Salts

Inorganic ions can be selectively processed by the MTs epithelium, stored as salts in the lumen, and excreted to perform various mechanical and physiological tasks. 

Harting [98], in 1873, described the occurrence of calcospherites in the lumen of the MTs of the Tephritidae *Acinia herachlei* Lioy, 1864 (now *Euleia heraclei* (Linnaeus, 1758)). Calcospherites are deposits of calcium carbonate corpuscles in an organic matrix, usually present in the fat cells of the fat body of dipteran larvae [99,168]. However, in *E. heraclei*, calcospherites are only found in the lumen of the MTs of the larvae, especially in their terminal segments, but not in the fat body [99]. During the last larval molt, calcospherites gradually dissolve in the perivisceral fluid, then the calcium carbonate moves to the ecdysial fluid passing through the newly formed cuticle of the pupa. Finally, when the ecdysial fluid is reabsorbed, calcium carbonate sediments on the internal surface of the puparium in a hard, friable layer. This process was defined by Keilin as “ecdysial elimination” [99]. Some microscopic (TEM, SEM) observations, X-ray crystallography, and chemical, analytical, and spectroscopic analyses of the MTs and puparia of *E. heraclei*, and other tephritid flies, would be necessary to support this apparently unique occurrence of calcospherites.

In the larvae of the Muscidae *Musca autumnalis* De Geer, 1776, numerous granules of minerals are formed and stored in the distal region of their pair of anterior MTs. Granules are smooth and spherical with a diameter of 0.2 to 10.0 µm. They are mainly composed of calcium, phosphorus, magnesium, and potassium with additional traces of sodium, manganese, iron, zinc, and copper arranged in several concentric layers and bonded by an organic matrix with a fibrous structure made of amino acids and carbohydrates [103,104,105]. Krueger and co-authors [105] suggest that the dissolution and release of these granules into the hemolymph happen in the proximal region of the MTs where the pH is more acid, as carbonate and phosphate salts of calcium and magnesium are insoluble in an alkaline environment. Therefore, *M. autumnalis* MTs display a regional specialization within each tubule [105]. Krueger et al. [106] also analyzed the larval and puparial hemolymph of *M. autumnalis* to verify how the calcium, magnesium, and phosphorus stored in the MTs are translocated to the pupal cuticle. The authors observed that a decrease in the mineral salts level in the hemolymph of the larva transforming into pupa corresponds to a proportional increase in the same elements in its puparial cuticle. The alkaline pH of the puparial cuticle is similar to that present in the MTs distal region, where the mineral granules are formed. Approximately 80% of the inorganic salts remain in the puparium left behind after the adult emergence, 5% are excreted in the meconium, and the remaining 15% are found in adults and their frass [106]. In an additional experiment, the calcium transport from the hemolymph to the puparium was successfully confirmed by rearing larvae on a radiolabeled medium containing the radioactive isotope ^45^Ca [106]. The finished puparium of *M. autumnalis* contains 62% of inorganic salts (calcium, magnesium, and phosphorous) and lipids, chitin, proteins, and uric acid as organic compounds [102]. A similar composition is also reported for the puparium of the close relative *Musca fergusoni* Johnston and Bancroft, 1920 [101]. The puparium of both species is brittle and white, not hard and dark brown as usual in Cyclorrhaphous flies, and those characteristics suggest that the calcification could occur as an alternative to the more common phenolic tanning to confer hardness [100]. Further investigations on the puparium of other species belonging to the genus *Musca* and, in general terms, of Cyclorrhapha flies, could corroborate such observations and clarify the role of calcium and/or tannins in the puparial cuticle hardening.

Another peculiar use of calcium in Diptera is displayed by the larvae of *Ephydra* (*Hydropyrus*) *hians* Say, 1830 (Diptera: Ephydridae). The median region of their pair of anterior MTs is modified into “lime glands” which contain nearly pure calcium carbonate. Calcium carbonate can regulate the high concentrations of environmental carbonate and bicarbonate through precipitation. Such mechanism allows the larvae of this fly to live in hypersaline, alkaline lakes [107]. 

The overwintering larvae of the Diptera Tephritidae *Eurosta solidaginis* (Fitch, 1855) need to acquire freeze tolerance during the Autumn. The third instar larvae begin this process in late September synthesizing first glycerol, then sorbitol, and trehalose. The further regulation of supercooling is possible thanks to the large crystalloid spheres found in the lumen of the anterior pair of their MTs. Each larva contains 24-45 spherules, 100 to 300 µm in diameter, with the surface made of numerous round particles. The main components of the spheres are calcium and phosphorus, in the form of tribasic calcium phosphate hydrate, with traces of magnesium. This compound is a heterogeneous ice-nucleator and acts as a nucleus for water crystallization, inducing ice formation across the tubule at higher sub-zero temperatures. The ice nucleation in the extracellular space can limit lethal intracellular freezing, thus promoting the survival of *E. solidaginis* larvae at very low winter temperatures [108].

In some females of the order Phasmatodea, the calcium carbonate is collected in the MTs, transferred to the hemolymph, and utilized to harden the eggs’ chorion. Phasmids are polynephric, and the Lonchodidae stick insect *Dixippus morosus* (Brunner von Wattenwyl, 1907) (now *Carausius morosus* Brunner von Wattenwyl, 1907) has about 24 superior tubules, 134 inferior “calciferous” tubules, and about 32 tubules of a third kind (defined “appendices” of the midgut) [111]. The tubules are arranged in 20 to 27 groups each one including five calciferous tubules and one excretory tubule [112]. The histochemical analysis of the *C. morosus* MTs showed the presence of uric acid in the superior tubules, as they perform regular excretion. In the distal region of the inferior ones, even if similar in shape, are found white granules with high levels of calcium carbonate and traces of magnesium, but only small amounts of uric acid. Such inferior tubules, present in females, are poorly developed in males and absent in nymphs [109]. Females discharge the excess magnesium, potassium, and sodium deriving from the diet through the feces, but they deposit the calcium of the inferior tubules in the eggshells. In this case, eggs constitute a second channel of elimination [111]. Ramsay [110] is praised for the development of the technique that, in 1954, allowed him to work for the first time with a singular, isolated tubule of *C. morosus* kept alive for some hours. Such technique, with minor changes, is still applied after about 70 years to study MTs and other fluid-secreting tubules [169].

The “appendices” of the midgut, investigated in 2014 in the females of *C. morosus* and some Phasmatidae, Heteropterygidae, and Pseudophasmatidae species, are thin coiled filaments connected to pear-shaped ampullae. “Appendices” cells are smaller and less rounded than the tubules cells and have shorter basal plasma membrane infoldings with only a few associated mitochondria. They are specialized in water and ion transport and alkalinisation of the midgut lumen [113], expel cations, sequester organic alkaloids, and regulate calcium [114], lipids, fatty acids, lipoproteins, and lipophilic substances such as some nutrients and xenobiotics [17]. This multidisciplinary approach (biochemical, morphological, physiological, genomic) would be welcomed if applied to the study of superior and inferior tubules too.

The Lepidoptera Erebidae *Leucoma* spp. and Lasiocampidae *Eriogaster* spp. and *Malacosoma* spp. build large, egg-shaped cocoons (Figure 7), from which the vulgar name “eggar moths” derives. The cocoons are very hard and from pale yellow to dark brown in color. They are made of silk produced by the labial glands, woven in a loose and open framework. The hardness is conferred by a paste of calcium produced by the larvae MTs [115,116] and incorporated (more or less evenly according to the species) around the silk threads [119]. For example, the main component of the crystalline powder covering the cocoon of *Malacosoma neustria* Linnaeus, 1758 is calcium oxalate monohydrate deriving from the oxalic acid in the diet [118], and the lemon-yellow pigment is assumed to be saturated after the crystallization inside the MTs lumen [117]. Some microscopic (TEM, SEM) observations, X-ray crystallography, and chemical, analytical, and spectroscopic analyses should be applied to the cocoons of such Erebidae and Lasiocampidae moths and other closely related species to verify their structure and the forms of calcium salts they include as well as their potential origin from the diet.

The inorganic salts processed by the MTs are also employed by some Cerambycidae beetles to cover their pupal cells walls and build the respective closing lids [86,121,122]. The *operculum* built by *Cerambyx* sp. larvae to close the entrance of their galleries dug in oaks trunks is, on the outside, a heap of woody debris made with chewed wood and, on the inside, a one-piece mineral block, as hard as limestone. This *operculum* is made of calcium carbonate blended with organic, albuminous cement, which gives it calcareous compactness [86]. Mayet [120] reported that, in *Cerambyx velutinus* Brullé, 1832 (now *Cerambyx welensii* (Küster, 1845)) larvae, four out of the six MTs are more developed and appear swollen and whitish because of the presence of the calcium carbonate. During the *operculum* construction, the whitish liquid produced by the MTs, containing the calcareous suspended granules, runs through the digestive tract of the larva (that is empty) and is then ejected from the mouth. The presence of limestone in the soil underneath the host plants is supposed to be essential for cerambycids to accumulate enough calcium carbonate to cover and close their hibernation galleries [120]. After more than a century, there are no updated observations about cerambycids and their peculiar usage of calcium carbonate, nor about the exact chemical composition of the organic, albuminous cement. In this case, an investigative approach to proteins, coupled with biological observations and chemical analyses, could lead to interesting new data. 

### 2.7. Bioluminescence

Another specialized function of the MTs is bioluminescence, a major tourist attraction displayed by some Australian and New Zealand glowworms, namely the larval stages of *Arachnocampa* spp. (Diptera: Keroplatidae) flies. Their colonies live in rainforests, humid, dark caves, ravines, gorges, tunnels, or bushes on banks [129,130], and these predator larvae build hollow tubular nests made of silk and mucus products originating from two distinct kinds of glands [125]. The nests are suspended from the cave ceiling by silk filaments, and larvae can glide backward and forward through them [126]. The small flying or fallen prey are captured by silk threads called “snares”, 1 to 50 cm in length, hanging down vertically from the nests (Figure 8). These fishing lines have sticky droplets of mucus on the surface, arranged at regular intervals [126,133]. When a prey remains ensnared, the keroplatid larva detects it thanks to chemo- and mechanoreceptors, then quickly pulls it up and eats it alive [129]. 

The lure for the prey is represented by a blue-green light (λ about 480 nm) emitted from specialized “lanterns” located at the posterior end of the body, derived exactly from modified MTs [123,127,128]. In *Arachnocampa (Arachnocampa) luminosa* (Skuse, 1891), adults and pupae of both sexes also glow. The females’ light is larger and brighter than that of the males and is probably used to attract the opposite sex. In female pupae, the light flashes brightly during the last days of development, while, in male pupae, the light becomes weak a few days before the emergence [126]. 

In *A. luminosa* larvae, the four MTs have four morphologically distinct regions, and their swollen distal tips form the “light organs”. The epithelial cells of the light organ are large, cuboidal, with a grainy, dense cytoplasm containing glycogen, small, circular nuclei, ribosomes, large mitochondria, and smooth endoplasmic reticulum. The ventral and lateral surfaces of the light organ are covered by a layer of densely packed, fine tracheoles that acts as a “reflector” to turn the light downward [123,124,127]. The blue-green glow of keroplatids results from a chemical reaction that involves a luciferin-luciferase system, magnesium-adenosine triphosphate (Mg-ATP), and molecular oxygen provided by the numerous ventral tracheoles [132]. Recently, it was demonstrated that the luciferase enzyme belongs to the same protein family found in the fireflies (Coleoptera: Lampyridae) [131,134], but the luciferin substrate is unexpectedly produced from tyrosine and xanthurenic acid and is completely different from the luciferins of any other glowing creature [134]. Moreover, in the “lanterns” were found enzymes of the AMP/CoA-ligase superfamily (luciferase-like enzymes) that might play an important role in the detoxification of xenobiotics [16]. 

### 2.8. Silk-like Fibers

Most insect species that spin cocoons use the silk produced by their labial silk glands. However, some insects belonging to the orders Coleoptera, Hymenoptera, Neuroptera, and Thysanoptera use silk-like fibers with a composition and texture very similar to silk but produced by the MTs and extruded from the anus. 

The first scientific evidence of silk-like fibers production dates to 1904, when Silvestri [135] reported that the MTs of the Coleoptera Carabidae *Lebia* (*Lebia*) *scapularis* (Geoffroy in Fourcroy, 1785) can secrete this material to build its cocoon. The first instar of this hypermetamorphic beetle is campodeiform, with well-developed mouthparts and long, agile legs. Due to the intense nutrition, the larva grows to become fusiform for the massive fat accumulation, and it is no longer able to easily move. If not properly protected, it could be easily preyed on by other predators. For this reason, it spins a cocoon for its protection and the defense of the following second larval stage, pre-pupal, and pupal stages. In the spinning larva, the proximal end of its four MTs is swollen and has walls made of glandular cells with granular cytoplasm vacuolated around the nuclei. In the lumen of this part of the tubule, only the silk-like compound was found [135]. This substance passes into the hindgut and is temporarily stored in the rectum which acts as a reservoir, while the pygidium works as a spinneret. The spinneret does not have specific appendices, but, being a narrow tube, all the material pushed out by the rectal musculature assumes the shape of a thread. The cocoon is oval-shaped and from citrine yellow to brown in color. It is built with filaments of variable thickness (5-40 µm), intertwined and fused irregularly, which create a loose cover that initially shows the larva, later the pre-pupa and pupa inside of it. Internally, the weaving is denser, and the fibers are thinner, so the inner surface of the cocoon seems covered by a smooth membrane [135]. To the best of our knowledge, this is the only report about this specialized function of MTs in *L. scapularis*.

The production of a silk-like fiber to build cocoons is a common feature also among other Coleoptera, namely the Curculionidae Hyperinae. The larvae of *Phytonomus arator* (Linnaeus, 1758) (now *Hypera arator* (Linnaeus, C., 1758)) have six MTs that produce the silk-like substance from their central part and a lubricant from the apical part. Using their mouthparts, the larvae gradually collect from the anus the silk-like fiber that hardens in a thick strand. This is used by the larva to produce its bright yellowish-green cocoon: the characteristic color is due to crystals (also found within the MTs) that encrust the transparent intertwined filaments. First, the outer wall of the cocoon is woven, and then its inner layer is smeared with the lubricant liquid that dries quickly in contact with the air. Each deposited layer of this fluid progressively closes the holes in the produced cocoon [136]. The exact composition of the two substances produced by the MTs and the cocoons’ structure could be investigated, respectively, through chemical analyses (e.g., colorimetric, chromatographic, spectrophotometric) and microscopy and X-ray crystallography techniques for a better understanding of this phenomenon. 

In *Hypera postica* (Gyllenhal, L., 1813) and *Hypera rumicis* (Linnaeus, C., 1758), a silk-like fiber is secreted by the proximal pinkish-brown region of their six MTs, where spherical granules that determine the characteristic color are present. The silk-like substance is mainly composed of alanine, glycine, proline, and serine. It is temporarily stored in the larval rectal sac and then poured out via the anus. The freshly produced protein is pale pinkish-brown then quickly darkens when in contact with air. Finished cocoons are brown and formed by a coarse weave (Figure 9a) [138].

In *Donus crinitus* Boheman, 1834, the MTs of the fourth instar larva produce a silk-like fiber utilized in the cocoon construction. The first three instars of this species are active, while the fourth one eats for a few days and then, when mature, enters a quiescent phase before spinning the cocoon. Larval MTs are, like in the other Hyperini previously reported, six: four longer and placed in the anterior part of the body and two shorter and placed in the posterior part. While the fourth instar is still active, a histological differentiation in the MTs becomes evident. In the proximal segment, epithelial cells are small while, in the distal segment, cells are large and contain nuclei and cytoplasm constantly growing in volume and granules increasing in number and dimensions. Afterward, when the larva becomes quiescent, the filled MTs occupy most of the hemocoel. In the mature larva, the distal part of its MTs is full of a fibrous, dense secretion used to build the cocoon and the proximal part discharges a mucoid product. The mixed secretions are poured into the gut and accumulate in the rectal ampulla, which works as a temporary reservoir before spinning [137]. Even the last instar larva of *Metadonus vuillefroyanus* (Capiomont, 1868) (now *Phytonomus vuillefroyanus* Capiomont, 1868 (provisionally accepted species)) builds a characteristic cocoon with two layers: the inner one consists of a silk-like mesh elaborated by the MTs, and the outer one contains soil particles (Figure 9b). The use of soil can be tactical to camouflage the cocoon and protect the pupa from adverse weather conditions. This behavior is unusual given that Hyperinae beetles usually pupate on or above the ground but do not employ soil particles [139].

Hymenoptera species usually pupate inside a cocoon made of silk; however, the parasitic wasps of the Eulophidae genus *Euplectrus* build their cocoons with a silk-like fiber secreted by the MTs [141,142]. In 1927, Thomsen [140] observed that *Euplectrus bicolor* (Swederus, 1975) larvae (Figure 9c) MTs produce a liquid material that coagulates in threads while exiting from the anus (Figure 9d). The cells of their MTs are large, highly vacuolated, and with highly ramified nuclei. The last four urites of the fourth instar larval body form a narrow tail-like projection, where the midgut internally ends. This tail-like projection is telescopic and movable like an arm and can fasten the liquid filament all around the mature larva, sewing the cocoon. The mesh is loose and coarse, and the silk-like fibers are extremely variable in thickness [140]. To this day, this specialized function of MTs is still considered a prerogative of the genus *Euplectrus* [143,144], apart from *E. josefernandezi* Hansson, 2015, but there are no updated investigations about the chemical composition of the silk-like fiber, cocoon’s structure, or other features.

In the larval stages of Neuroptera Planipennia species, the midgut and hindgut are not connected. Therefore, feces are accumulated in the midgut until adulthood, and the hindgut stores and expels the products originating from the MTs [58,170]. During larval development, these substances include the adhesive and defensive secretions reported in Section 2.3 with reference to some chrysopids. Later, as pupation approaches, MTs undergo a drastic histological change in about two days (cells become swollen and with ramified nuclei), ceasing their osmoregulatory role and starting to secrete a silk-like fiber with different purposes. In some genera, almost the entire length of the MTs is involved in this secretion, while in others only the posterior two-thirds of the MTs assume this function [58,153]. 

In 1941, Jones [145] observed a Chrysopidae aphis-lion larva producing a semi-fluid silk-like substance coming from the spinneret placed at the tip of its abdomen. With back-and-forth movements of the body, this fluid comes out and quickly dries into hard strands that are used to build a debris packet fastening snail shells on the back. Sometimes larvae also collect live snails that, subsequently, will be eaten [145]. *Leucochrysa* (*Leucochrysa*) *insularis* (Walker, 1853) carries on its back several snail species and other insects’ body parts [146].

When the third instar larvae of the Chrysopidae *Chrysopa perla* (Linnaeus, 1758) are mature, they spin white, oval-shaped cocoons the size of a green pea. They use the silk-like fiber produced by the eight MTs, stored in the rectum and then extruded from a telescopic spinneret. While spinning, the anterior tubules cells are wider and thicker, with large, branched nuclei, and the granular cytoplasm becomes highly vacuolated. Cocoons are placed inside rolled leaves, under leaves fallen on the ground or in the soil, where the sand grains can become part of the external wall [147,150]. In the pre-pupa of the closely related species *Chrysoperla comanche* (Banks, 1938), the hindgut epithelium appears enlarged, made of cells with long microvilli involved in the secretion and protruding in the intestinal lumen. In the gut’s lumen, however, are present two substances: one is epithelium-associated (defined as cuticulin silk by Rudall and Kenchington [171]) and employed to build the solid inner layer of the cocoon, while the other one, proteinaceous in nature, is elaborated by the MTs and used in the external fibrous wall [58]. 

A double structure was also observed in the cocoon of the Chrysopidae *Mallada signatus* (Schneider, 1851). The outer layer is made of loosely woven silk-like threads with a diameter of about 2.0 μm, while the inner layer consists of similar fibers embedded in a lipidic matrix. The main constituent of the cocoon is a single small protein (MalFibroin, 49 kDa) that provides a scaffold for the lipid layer. MalFibroin is composed of alanine, asparagine, and glycine. The protein gives mechanical support and makes the cocoon more protective, and the lipids, being hydrophobic, prevent pupal desiccation. The fibers can also display adhesive properties useful to anchor the cocoon to the substrate and stick leaves and debris on it for camouflage [155]. The chemical, chromatographic, spectroscopic, microscopic, and proteomic analysis of the different substances produced by the MTs in the before mentioned chrysopids could clarify if they have the same composition and structure and whether there are similarities with the MalFibroin discovered in the cocoon of *M. signatus.*


Meinert [148] first observed that the rectal sac of some Neuroptera Myrmeleontidae larvae is a reservoir of a product produced by the MTs [148]. The larvae of *Myrmeleon (Myrmeleon) formicarius* Linnaeus, 1767, active predators of ants, stop eating a few days before pupation and start digging deeply in sandy soils. At that moment, the proximal part of their eight MTs undergoes some histological changes that proceed distally. Cells become inflated and irregular and the nucleus large and branched. The silk-like fiber deriving from the MTs is stored in the rectum and then passes through two telescopic spinnerets, surrounded by muscles [151]. First, the larva spins the outer wall of the cocoon with a loose strand that incorporates sand grains (Figure 9e) [152]; later, it covers the inner wall with a shiny, white layer of densely woven fibers [151]. Before pupation, the MTs of *Myrmeleon* (*Myrmeleon*) *uniformis* Navás, 1920 larvae cease their osmoregulatory role and start to synthesize the silk-like fiber. The number of nucleolar bodies in the MTs increases with a consequent increment in the synthetic activity, and the most abundant protein detected through electrophoresis is a precursor of silk fibers with a molecular weight of about 32–43 kDa [157].

The larva of the Neuroptera *Sisyra umbrata* Needham in Needham and Betten, 1901 (now *Sisyra vicaria* (Walker, 1853)) (Sisyridae) has piercing-sucking mouthparts and feeds on freshwater sponges. Its MTs are modified in their middle proximal portions to produce a silk-like fiber. The secreting cells are larger and more irregular in shape than the usual MTs cells [149]. They display singular ramified nuclei like the proper silk gland cells of caterpillars [172]. The receptacle and outlet of the secretion is a straight tube, probably a modification of the small intestine, connected to the spinneret formed by the tenth abdominal segment. The walls of the spinneret are surrounded by strong, circular muscles that aid the ejection of the silk-like substance when the construction of the cocoon starts [149]. Silk-like cocoons were also observed in other *Sisyra* species [156], but their exact composition is still unknown.

Larvae of Neuroptera Ascalaphidae species are ground and litter dwellers. *Ascalaphus, Ascaloptynx*, and *Ululodes* spp. larvae are sedentary predators that remain motionless for long periods, keeping their jaws spread open and relying on camouflage to ambush prey. The third-age larvae use a retractile spinneret, positioned on their tenth abdominal segments, to spin a silk-like compound elaborated by the MTs. The silk-like fluid quickly dries in a strand when in contact with air, and it is employed to fasten together loose particles of sand and debris. The building of the cocoon starts with a ring of woven soil that surrounds the larva. Then, the walls are erected through elongations and contortions of the larva’s abdomen, legs, and jaws. The completed structure is spherical and consists of an outer sand-silk matrix and an inner layer of the thickly spun silk-like fiber [154,158], but the protein structure was not investigated. 

Thysanoptera species, even if defined heterometabolous (insects that undergo incomplete metamorphosis), develop through two active and mobile larval stages and two or three aphagous and quiescent stages: the propupa and pupa. In some species, the second-age nymph creates a silk-like cocoon into which it pupates. This behavior was firstly reported in *Aeolothrips fasciatus* (Linnaeus, 1758) [159] and later in other species of the Aeolothripidae (*Aeolothrips* sp. and *Franklinothrips* sp.) [160,162], Heterothripidae (*Aulacothrips* sp.), Melanthripidae (*Ankothrips* sp.), and Thripidae (*Anaphothrips* sp. and *Odontothrips* sp.) families [161,163,164]. 

Cocoons are built with an anal excretion, whose origin from the MTs was assessed in 2010. Conti and co-authors [166] documented the ultrastructural modifications of the MTs’ epithelium during pupation in the Aeolothripidae *Aeolothrips intermedius* Bagnall, 1934 (Figure 10a). Mature larvae seek a suitable place to spin their cocoons, generally at the bifurcation of two veins of the leaf or in soil cracks [165]. Then, they frantically shake the abdomen from right to left and vice versa to stretch the thread coming out from the anus. 

At first, the larva weaves a larger mesh net (Figure 10b) to anchor and protect the future cocoon. At the end of the spinning, the cocoon displays a double covering: an outer, loose layer and an inner, dense one (Figure 10c). SEM microscopic observation revealed a net of filaments of different diameters. Moreover, most of the fibers have a smooth surface, while some have small spines (Figure 10d) [165]. The spectrometric analysis of the silk-like fiber showed a copious presence of β-*N*-acetyl-glucosamine, the same secondary amide found in Neuroptera and some Coleoptera cocoons (Rudall and Kenchington, 1971). When the larva spins the cocoon, the cytoplasm of its two MTs epithelial cells appears rich in the rough endoplasmic reticulum and contains numerous Golgi bodies involved in a highly intense secretion activity, as testified by their copious production. In fact, the lumen of the larval MTs appears filled with an electron-dense secretion (Figure 9f) that then migrates towards the terminal part of the MTs [166]. Similar investigations on the previously cited Thysanoptera species and additional ones could support the outdated observations and underline potential similarities between the structure and composition of their cocoons and those of Coleoptera, Hymenoptera, and Neuroptera specimens.

## 3. Conclusions

In conclusion, this review highlights the vast functional diversity of insects’ MTs that, besides the conserved role of excretory and osmoregulatory organs, have assumed an extraordinary number of specialized functions (Table 1) scattered along the whole evolutionary range of the Insecta. Although many studies on these peculiar organs have been performed between the 1850s and 1920s, they were not followed by up-to-date experiments that would have shed light on additional aspects of insect physiology, biology, and behavior. For example, the histochemical changes in the MTs of the clastopterid and machaerotid nymphs and the composition of the mucofibrils and proteins there produced were extensively studied by Marshall and coworkers in the 1960s and 1970s, but no other studies have been performed since then. New details about this specialized function, perhaps through the quantification and qualification of the proteins produced, could bring significant starting points for the classification of the Clastopteridae and Machaerotidae families, whose phylogeny and taxonomy are still under debate. 

Similarly, a new investigative approach to proteins, coupled with biological observations and chemical and microscopic analyses, could also be useful to validate the information about the adhesive secretions produced by some chrysomelid beetles MTs, sugary and proteinaceous substances produced by some cicada nymphs MTs, the glycoproteinaceous and lipoproteinaceous matrix secreted by specialized cells of the MTs of some scale insects, the calcium carbonate and the organic, albuminous cement produced by some cerambycid beetles MTs, and the silk-like compound used to spin the cocoon by some Coleoptera, Hymenoptera, Neuroptera, and Thysanoptera species. The modern, proteomic technologies could also determine if the adhesive substance produced by some chrysopid larval instars is the same silk-like compound later used to build the cocoon or a precursor.

Moving to the inorganic salts, the apparently unique occurrence of calcospherites in the MTs of *E. heraclei*, their exact composition, and the composition of the puparium of this tephritid fly would require new microscopic (TEM, SEM) observations, X-ray crystallography, and chemical, analytical, and spectroscopic analyses to be confirmed. Such techniques could also be applied to the cocoons of erebid and lasiocampid moths to verify their structure and the forms of calcium salts they include as well as their potential origin from the diet. 

On the other hand, some of the non-excretive functions of the MTs still arouse the scientific community’s interest. For example, many recent studies focused on the composition and possible functions of the foam produced by the proximal and distal segments of the MTs of aphrophorid and cercopid true spittlebugs. The possible functions of the “appendices of the midgut”, one kind of MTs present in stick insects, were recently clarified through transcriptome analysis. Even the biochemical reactions involved in the stunning phenomenon of bioluminescence, occurring in the “lanterns” of some keroplatids MTs, were recently investigated through chromatography, mass spectrometry, spectroscopy, transcriptome sequencing, proteomic, and genomic. Some interesting potential applications of novel enzymatic systems may result from studies on some of the non-excretive functions of the MTs through the new proteomic and genomic technologies such as high-throughput transcriptome sequencing.

## Figures and Tables

**Figure 1 insects-13-01001-f001:**
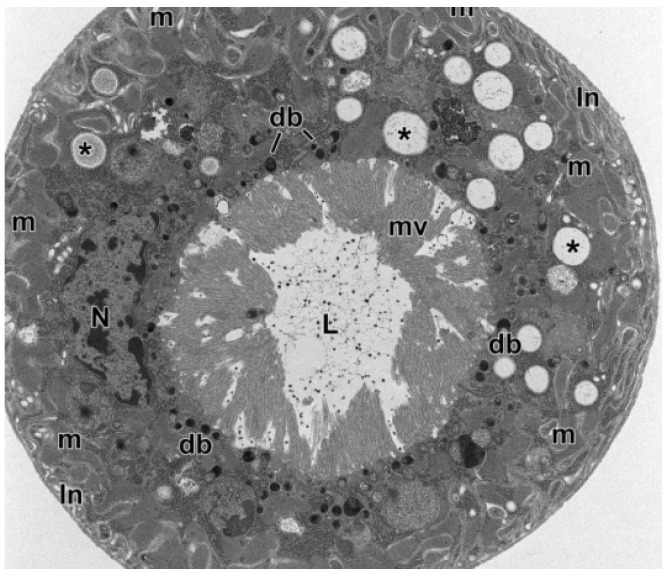
Cross-section through a Malpighian Tubule (MT) of an *Aeolothrips intermedius* Bagnall, 1934 (Thysanoptera: Aeolothripidae) adult. The epithelial cells show a basal system of membrane infoldings with associated mitochondria. The nucleus occupies the central cell region. In the cytoplasm, there are numerous electron-transparent inclusions, sometimes with concentric layers (asterisks). Electron-dense bodies are visible at the apex of epithelial cells, at the base of microvilli. In the MT lumen are present scattered small electron-dense granules. db = electron-dense bodies, In = membrane infoldings, L = lumen, m = mitochondria, mv = microvilli, N = nucleus. From Conti et al. [5].

**Figure 2 insects-13-01001-f002:**
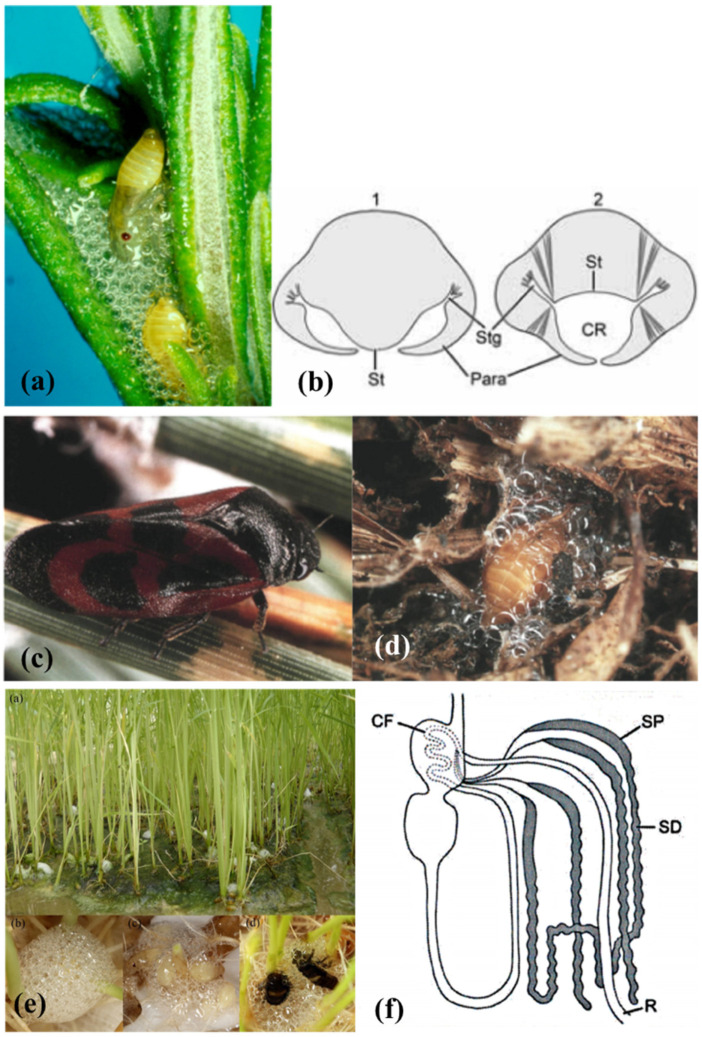
(**a**) *Philaenus spumarius* (Linné, 1758) nymphs inside the foam nest. (**b**) representation of the abdominal morphology of true spittlebugs nymphs during the production of the foam, in correspondence to the first (1) and fourth (2) urite. St = sternum, Stg = stigmas, Para = paratergal lobes around the respiratory cavity (CR). From Conti et al. [5]. (**c**) *Haematoloma dorsata* (Ahrens, 1812) adult and (**d**) nymph in its foam nest. Images courtesy of José María Cobos, Ministerio de Agricultura, Pesca y Alimentación C/Almagro, Madrid, Spain. (**e**) *Callitettix versicolor* (Fabricius, 1794) spittle masses produced by the nymphs at the roots of rice (*Oryza sativa* L. Poaceae) plants. Images courtesy of Ai-Ping Liang and Xu Chen, Institute of Zoology, Chinese Academy of Sciences, Beijing, China. (**f**) Representation of the gut complex, including the Malpighian Tubules (MTs), of Homoptera Cercopoidea preimaginal instars. CF = filter chamber, SP = proximal segment of the MTs, SD = distal segment of the MTs, R = rectum. From Conti et al. [5].

**Figure 3 insects-13-01001-f003:**
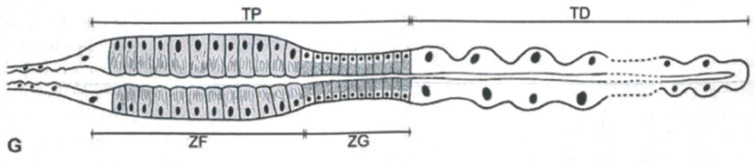
Representation of Machaerotidae nymphs Malpighian Tubules. TP = proximal tract, TD = distal tract, ZF = zone rich in fibrils, ZG = zone rich in granules. From Conti et al. [5].

**Figure 4 insects-13-01001-f004:**
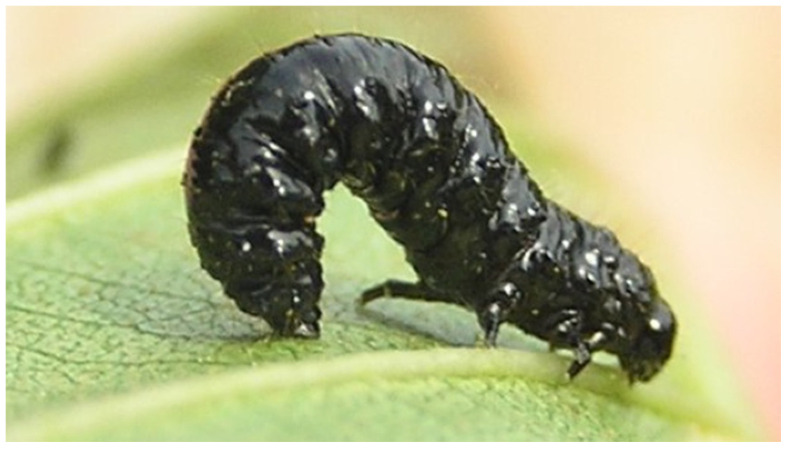
*Agelastica alni* (Linnaeus, 1758) larva. Image credit: I. Beentree.

**Figure 5 insects-13-01001-f005:**
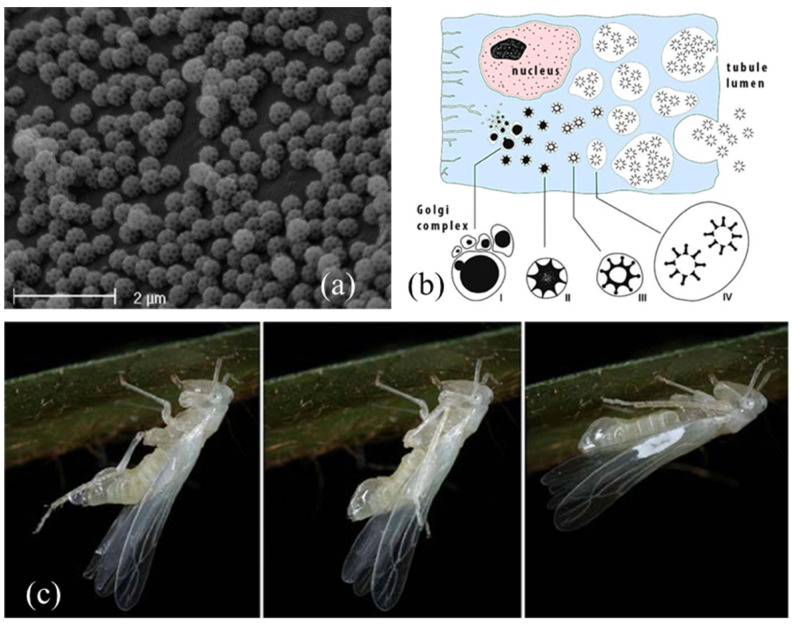
(**a**) Integumental brochosomes. Image courtesy of Andrea Lucchi, Department of Agriculture, Food and Environment, University of Pisa, Italy. (**b**) Development of brochosomes through four stages in Golgi-derived vesicles. Image credit: Roman Rakitov, licence CC BY-SA 4.0. ©2016 https://www.wikiwand.com/en/Brochosome#Media/File:Brochosome_secretion_En.jpg. (**c**) A freshly molted female of *Igutettix oculatus* (Lindberg, 1929) using hind tibiae to transfer the fluid medium containing brochosomes from the anus (Left) onto the forewings (Center) where it dries in white spots (Right). Images credit: Roman Rakitov, CC BY-SA 3.0. ©2009 https://www.wikiwand.com/en/Brochosome#Media/File:Igutettix_anointing.jpg.

**Figure 6 insects-13-01001-f006:**
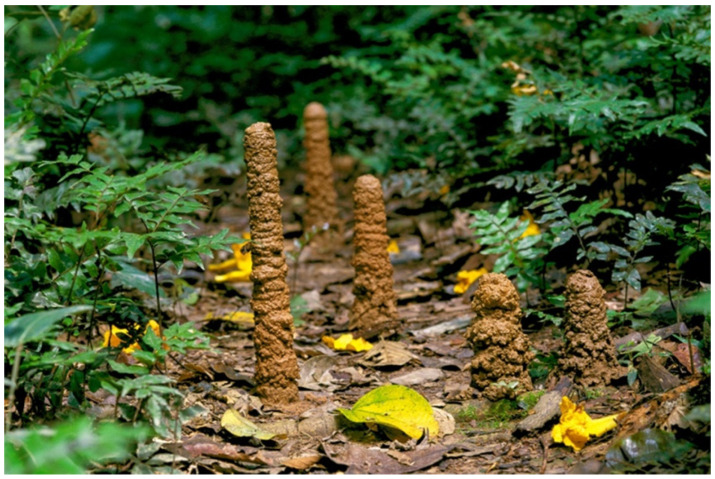
Exit tunnels built by *Fidicina chlorogena* Walker, 1850 (now *Guyalna chlorogena* (Walker, 1850)). Image courtesy of André Bärtschi, Vaduz, Liechtenstein.

**Figure 7 insects-13-01001-f007:**
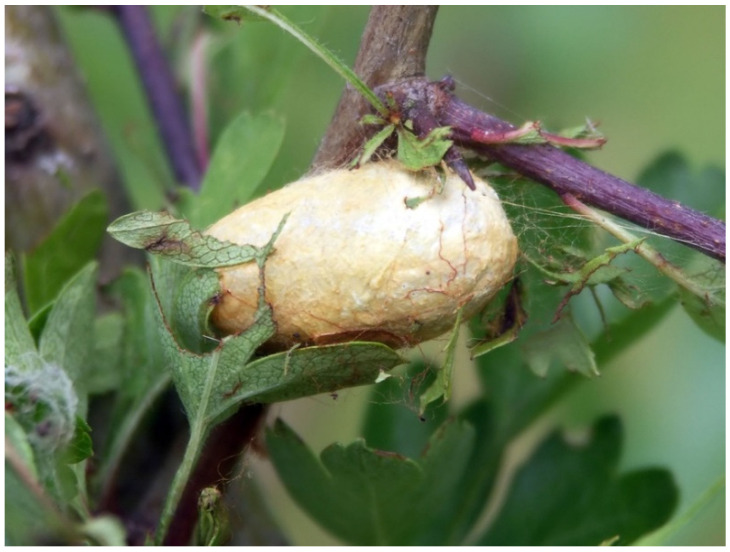
*Eriogaster lanestris* Linnaeus, 1758 egg-shaped cocoon. Image credit: Krzysztof Jonko. From Conti et al. [5].

**Figure 8 insects-13-01001-f008:**
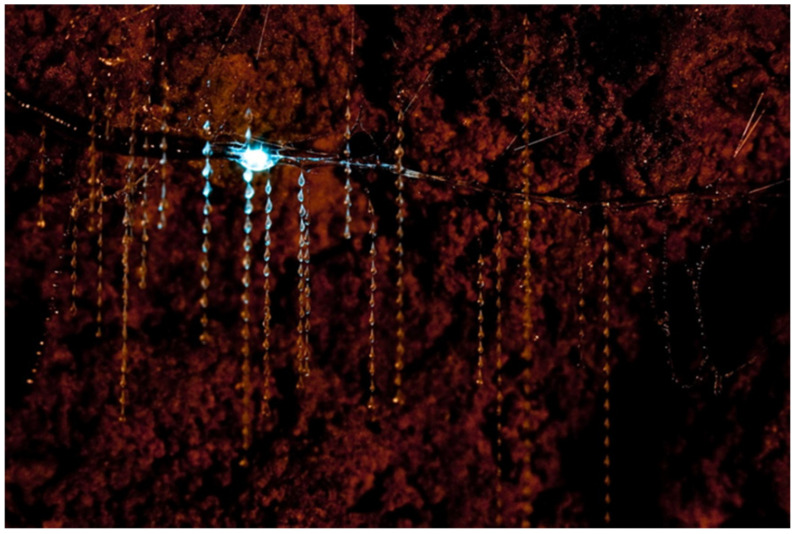
*Arachnocampa (Arachnocampa) luminosa* (Skuse, 1891) larva in its tubular nest from which hang down numerous silk threads with regularly arranged sticky droplets. Image courtesy of Chris Wills. From Conti et al. [5].

**Figure 9 insects-13-01001-f009:**
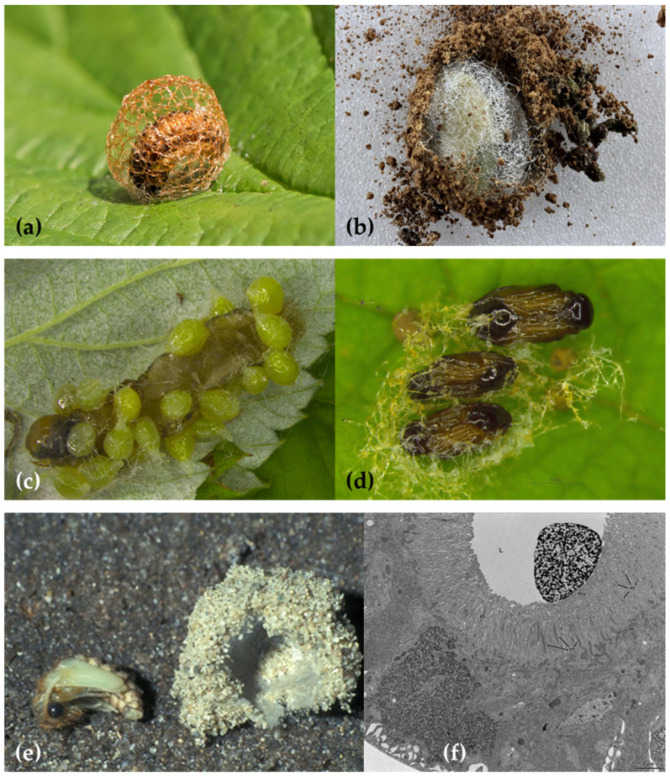
(**a**) *Hypera rumicis* (Linnaeus, C., 1758) cocoon. Image credit: Charles J. Sharp, CC BY-SA 4.0. ©2016 https://commons.wikimedia.org/w/index.php?curid=50017118. (**b**) *Phytonomus vuillefroyanus* Capiomont, 1868 cocoon. Image courtesy of Petr Bogusch, University of Hradec Králové, Czech Republic. (**c**) *Euplectrus bicolor* (Swederus, 1795) larvae on the caterpillar of *Orthosia* sp. (Lepidoptera: Noctuidae). (**d**) pupae with threads made of a silk-like fiber. Images courtesy of Špela Modic, Agricultural Institute of Slovenia, Ljubljana, Slovenia. (**e**) *Myrmeleon (Myrmeleon) formicarius* Linnaeus, 1767 pupa separated from its cocoon made of silk-like fibers and sand. Image credit: Saxifraga Foundation, CC BY-NC. (**f**) *Aeolothrips intermedius* Bagnall, 1934 s-instar larva cross-section of a Malpighian Tubule (MT). In the lumen is visible a large mass of an electron-dense secretion. From Conti et al. [5].

**Figure 10 insects-13-01001-f010:**
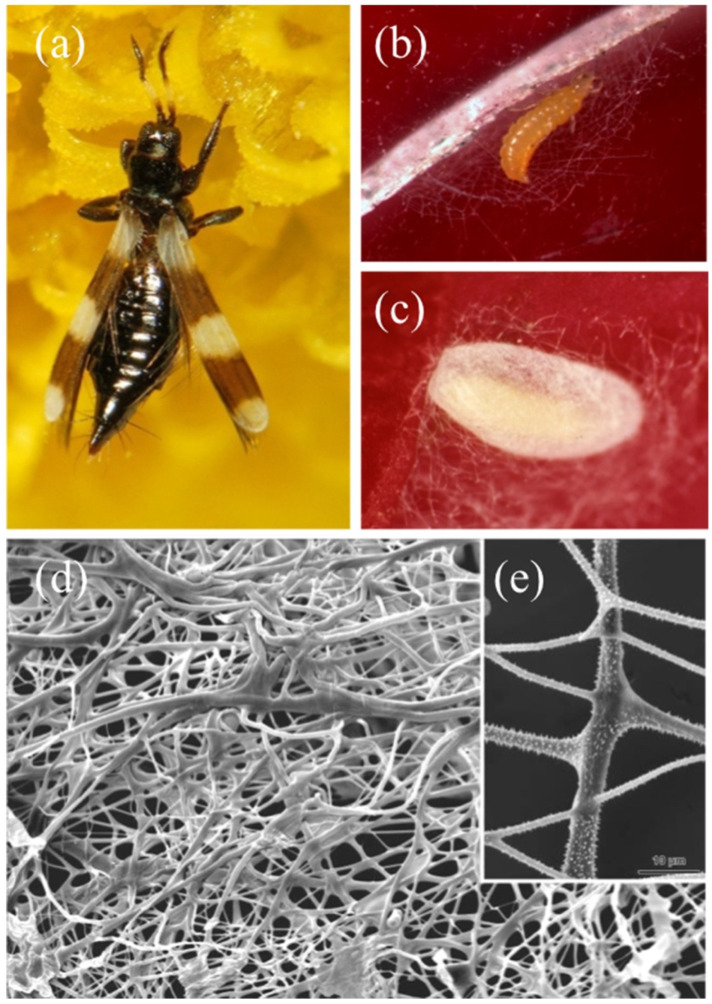
*Aeolothrips intermedius* Bagnall, 1934. (**a**) female. (**b**) mature larva at the beginning of the spinning procedure. (**c**) cocoon. (**d**) SEM detail of the cocoon fibers with (**e**) a close-up of the spines. From Conti et al. [5].

## Data Availability

Data are available on request from the corresponding authors.

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
