# Peer review of "Multiple Functions of Malpighian Tubules in Insects: A Review"

_insects, 2022, doi:10.3390/insects13111001_

Round 1

Reviewer 1 Report

This article reviews studies related to Malpighian tubules in a very systematic way. Not only the morphological structure of Malpighian tubules was analyzed, but also various biological functions of Malpighian tubules were summarized, including the production, processing, and storage of mucopolysaccharides, proteins, mucofibrils, adhesive secretions, brochosomes, silk-like fibers, and inorganic salts as well as the remarkable phenomenon of bioluminescence. Based on the review of previous research results, this paper analyzed some experimental results that are outdated and still need to be verified by advanced technology. In general, this systematic review points out the direction for future research and is very meaningful work. 

Author Response

We sincerely thank Reviewer 1, we really appreciate the comments on our manuscript.

Reviewer 2 Report

General comments:

This mid-length review focuses on the roles of Malpighian tubules in processes other than excretion and osmoregulation. These other processes have received much less attention and this review thus fills in a gap in the literature. The authors have collected information from a diverse range of 19th, 20th and 21st century studies. In some senses, the review is an appeal for more modern studies using transcriptomic and proteomic approaches to clarify the nature of the molecules included in the organic and inorganic secretions of Malpighian tubule cells. In view of biomaterial research on animal silks, studies highlighting the roles of the Malpighian tubules in silk production may have implications for development of biomimetics.

Specific comments:

Title: slightly clumsy wording. I suggest ‘Multiple functions of Malpighian tubules in insects: a review.’ As an alternative.

Line 7: change ‘as the main excretory organs in most insects’ to as the main excretory and osmoregulatory organs in most insects’

Line 37: change ; ‘their first discoverer’ to  ‘their discoverer’

Line 38: change ‘inserted at the midgut-hindgut junction’ to’ emptying their secretions into the gut at the midgut-hindgut junction’

Line 51: change ‘enhancing the ions and fluid transport” to ‘enhancing the capacity for ion and fluid transport’

Line 92: ‘Furthermore, MTs can also contrast the toxic activity…’ . ‘contrast’ is not an appropriate word. Perhaps change to ‘reduce’ or ‘mitigate’

Line 155: change ‘proteins secreting and exporting cells’ to ‘protein-secreting and -exporting cells’

Line 168: change ‘and their nutrition on xylem sap’ to ‘and their diet of xylem sap’

Line 174: meaning of  ‘and similar to that of the soil.’ is unclear.

Lines 207-218: I suggest re-arranging and re-wording slightly, as follows: ‘Their four MTs consist of three distinct regions: a lobulated distal segment, five times longer than the adjacent smooth proximal segment, and a slim outlet duct. A similar regional differentiation in the MTs is also reported in true spittlebugs ([34] – paragraph 2.1) and cicadas ([88] – paragraph 2.5). The proximal segment can be divided into two further zones: an anterior one with fibril-rich cells (fibril zone) that occupies three-quarters of the total length of the proximal segment,  and a posterior one with granules rich cells (granule zone) (Figure 3). The granule zone is responsible for foam production, …’

Line 222: ‘The granule zone is cytologically comparable to the entire proximal segment of true spittlebugs’ MTs, but it is shorter, as its function is not continuous over time [47].” I am not sure why the granule zone is shorter just because its function is not continuous over time.

Line 340: ‘ …while the inner covering helps to maintain the chamber clean from the excrements [80].’ I suggest re-wording: ‘ …while the inner covering helps to maintain the chamber free of excrement [80].’

Line 357: ‘The constant research…’ change to ‘Continual research….’

Line 358: ‘has satisfactorily highlighted’ change to ‘has examined in detail’

Line 371: ‘mechanical ad physiological properties’ to ‘mechanical and physiological properties’

Line 412: parallel, not parallely

Line 428: It is unclear what you mean by ‘As this phenomenon appears to be an exception…”. Exception to what?

Line 438 and elsewhere: ‘co-authors’ not ‘co-Authors’

Line 444: ‘The authors…’ not ‘The Authors….’

Line 458: suggest changing ‘could occur as a substitute of the more common phenolic tanning’ to ‘could occur as an alternative to the more common phenolic tanning’

Line 494: ‘Ramsay is praised for ……t other fluid-secreting tubules [169].’ This sentence does not really relate to the previous sentences and seems out of place.

Line 586: ‘A large part of the insects…’ change to ‘Most insect species…..’

Line 640-642: ‘In the lumen of the MT is visible a mass of electro-dense secretions used in the spinning of the cocoon. gl = glycogen, L = lumen, m = mitochondria, mv = microvilli, s = electron-dense secretions.’. The labels indicated in the figure caption are missing from the figure.

Line 650: change ‘gets evident’ to ‘becomes evident’

Line 653: not sure what you mean by ‘fulfilled MTs’

Line 683: I don’t understand what is meant by ‘ceasing the osmoregulation’.

Line 713: change ‘avoid the pupa desiccation’ to ‘prevent pupal desiccation’

Line 730: change ‘larvae stop to serve the osmoregulation’ to ‘larvae cease their osmoregulatory role’

Line 746:  change ‘The Neuroptera Ascalaphidae species have ground and litter dwellers larvae.’ to  ‘Larvae of Neuroptera Ascalaphidae species are ground and litter dwellers.’

Line 757: heterometabolous is not a very common term. Perhaps define in brackets. Or do you mean hemimetabolous?

Line 793: change ‘conserved role of excretory organs’ to ‘‘conserved roles of excretory and osmoregulatory organs’

Author Response

We thank the reviewer for the specific comments and general opinion on our review.

Specific comments:

Title: slightly clumsy wording. I suggest ‘Multiple functions of Malpighian tubules in insects: a review.’ As an alternative.

We changed the title as suggested.

Line 7: change ‘as the main excretory organs in most insects’ to ‘as the main excretory and osmoregulatory organs in most insects’

We added “osmoregulatory”.

Line 37: change ; ‘their first discoverer’ to  ‘their discoverer’

We deleted “first”.

Line 38: change ‘inserted at the midgut-hindgut junction’ to’ emptying their secretions into the gut at the midgut-hindgut junction’

We changed it as suggested.

Line 51: change ‘enhancing the ions and fluid transport” to ‘enhancing the capacity for ion and fluid transport’

We changed it as suggested.

Line 92: ‘Furthermore, MTs can also contrast the toxic activity…’ . ‘contrast’ is not an appropriate word. Perhaps change to ‘reduce’ or ‘mitigate’

Thank you for the suggestion, we used the verb mitigate.

Line 155: change ‘proteins secreting and exporting cells’ to ‘protein-secreting and -exporting cells’

We changed it.

Line 168: change ‘and their nutrition on xylem sap’ to ‘and their diet of xylem sap’

We changed the word “nutrition”.

Line 174: meaning of ‘and similar to that of the soil.’ is unclear.

We rephrased the sentence.

Lines 207-218: I suggest re-arranging and re-wording slightly, as follows: ‘Their four MTs consist of three distinct regions: a lobulated distal segment, five times longer than the adjacent smooth proximal segment, and a slim outlet duct. A similar regional differentiation in the MTs is also reported in true spittlebugs ([34] – paragraph 2.1) and cicadas ([88] – paragraph 2.5). The proximal segment can be divided into two further zones: an anterior one with fibril-rich cells (fibril zone) that occupies three-quarters of the total length of the proximal segment, and a posterior one with granules rich cells (granule zone) (Figure 3). The granule zone is responsible for foam production, …’

We changed the sentences as suggested.

Line 222: ‘The granule zone is cytologically comparable to the entire proximal segment of true spittlebugs’ MTs, but it is shorter, as its function is not continuous over time [47].” I am not sure why the granule zone is shorter just because its function is not continuous over time.

We rephrased the sentence.

Line 340: ‘ …while the inner covering helps to maintain the chamber clean from the excrements [80].’ I suggest re-wording: ‘ …while the inner covering helps to maintain the chamber free of excrement [80].’

We changed it as suggested.

Line 357: ‘The constant research…’ change to ‘Continual research….’

We changed the word “constant”.

Line 358: ‘has satisfactorily highlighted’ change to ‘has examined in detail’

We changed it.

Line 371: ‘mechanical ad physiological properties’ to ‘mechanical and physiological properties’

We changed it.

Line 412: parallel, not parallely

We changed it.

Line 428: It is unclear what you mean by ‘As this phenomenon appears to be an exception…”. Exception to what?

We are sorry, the sentence was misleading; we rephrased it.

Line 438 and elsewhere: ‘co-authors’ not ‘co-Authors’

We changed it throughout the text.

Line 444: ‘The authors…’ not ‘The Authors….’

We changed it throughout the text.

Line 458: suggest changing ‘could occur as a substitute of the more common phenolic tanning’ to ‘could occur as an alternative to the more common phenolic tanning’

We changed it as suggested.

Line 494: ‘Ramsay is praised for ……t other fluid-secreting tubules [169].’ This sentence does not really relate to the previous sentences and seems out of place.

We rephrased the sentence. We aimed to point out that a technique developed in 1954 is still applied in various studies.

Line 586: ‘A large part of the insects…’ change to ‘Most insect species…..’

We changed it as suggested.

Line 640-642: ‘In the lumen of the MT is visible a mass of electro-dense secretions used in the spinning of the cocoon. gl = glycogen, L = lumen, m = mitochondria, mv = microvilli, s = electron-dense secretions.’. The labels indicated in the figure caption are missing from the figure.

We are sorry. A part of the caption of Figure 1 was wrongly inserted under Figure 9. The caption is now corrected.

Line 650: change ‘gets evident’ to ‘becomes evident’

We changed it.

Line 653: not sure what you mean by ‘fulfilled MTs’

We are sorry, changed in “filled”.

Line 683: I don’t understand what is meant by ‘ceasing the osmoregulation’.

We rephrased it according to the suggestion about line 730.

Line 713: change ‘avoid the pupa desiccation’ to ‘prevent pupal desiccation’

We changed it as suggested.

Line 730: change ‘larvae stop to serve the osmoregulation’ to ‘larvae cease their osmoregulatory role’.

We changed it as suggested.

Line 746:  change ‘The Neuroptera Ascalaphidae species have ground and litter dwellers larvae.’ to  ‘Larvae of Neuroptera Ascalaphidae species are ground and litter dwellers.’

Thank you for the suggestion, we changed it.

Line 757: heterometabolous is not a very common term. Perhaps define in brackets. Or do you mean hemimetabolous?

We rephrased the sentence and defined the term heterometabolous in brackets.

Line 793: change ‘conserved role of excretory organs’ to ‘‘conserved roles of excretory and osmoregulatory organs’

We added “and osmoregulatory”.